# Early Motor Development Predicts Clinical Outcomes of Siblings at High-Risk for Autism: Insight from an Innovative Motion-Tracking Technology

**DOI:** 10.3390/brainsci10060379

**Published:** 2020-06-16

**Authors:** Angela Caruso, Letizia Gila, Francesca Fulceri, Tommaso Salvitti, Martina Micai, Walter Baccinelli, Maria Bulgheroni, Maria Luisa Scattoni

**Affiliations:** 1Research Coordination and Support Service, Istituto Superiore di Sanità, Viale Regina Elena 299, 00161 Roma, Italy; angela.caruso@iss.it (A.C.); letizia.gila@iss.it (L.G.); francesca.fulceri@iss.it (F.F.); tommaso.salvitti@iss.it (T.S.); martina.micai@iss.it (M.M.); 2Ab.Acus srl, Milano, via F. Caracciolo 77, 20155 Milano, Italy; walterbaccinelli@ab-acus.eu (W.B.); mariabulgheroni@ab-acus.com (M.B.)

**Keywords:** neurodevelopmental disorders, autism spectrum disorder, high-risk infants, motor development, screening, technology

## Abstract

Atypical motor patterns are potential early markers and predictors of later diagnosis of Autism Spectrum Disorder (ASD). This study aimed to investigate the early motor trajectories of infants at high-risk (HR) of ASD through MOVIDEA, a semi-automatic software developed to analyze 2D and 3D videos and provide objective kinematic features of their movements. MOVIDEA was developed within the Italian Network for early detection of Autism Spectrum Disorder (NIDA Network), which is currently coordinating the most extensive surveillance program for infants at risk for neurodevelopmental disorders (NDDs). MOVIDEA was applied to video recordings of 53 low-risk (LR; siblings of typically developing children) and 50 HR infants’ spontaneous movements collected at 10 days and 6, 12, 18, and 24 weeks. Participants were grouped based on their clinical outcome (18 HR received an NDD diagnosis, 32 HR and 53 LR were typically developing). Results revealed that early developmental trajectories of specific motor parameters were different in HR infants later diagnosed with NDDs from those of infants developing typically. Since MOVIDEA was useful in the association of quantitative measures with specific early motor patterns, it should be applied to the early detection of ASD/NDD markers.

## 1. Introduction

Neurodevelopmental disorders (NDDs) are a group of conditions with onset in the developmental period characterized by impairments of personal, social, and academic functioning (Diagnostic and statistical manual of mental disorders, DSM-5, APA 2013). The global rate of NDDs from 2009–2011 to 2015–2017 increased from 16.2% to 17.8% (e.g., Autism Spectrum Disorder (ASD): 1.1–2.5%; Attention-Deficit/Hyperactivity Disorder: 8.5–9.5%) [1]. Given the importance of early detection and intervention programs in improving infants’ and parents’ outcomes, previous research focused on the identification of early biomarkers and behavioral measures to determine risk status even before the emergence of clear behavioral symptoms [2,3]. In this scientific framework, the Italian National of Health hosts the Network for early detection of Autism Spectrum Disorders (NIDA Network), including pediatric hospitals and clinical research centers of Italian territory with high expertise in early detection and intervention programs in ASD. The NIDA Network delivered new tools and standards for research and clinical development across the nation and is currently coordinating the most extensive surveillance program of development in infants at risk for NDDs in Italy. The aim of the NIDA Network is the early detection of behavioral markers of NDDs to provide intervention programs in a timely manner.

Several studies revealed that some early behavioral measures might be effective in predicting abnormal developmental trajectories [4,5,6,7]. Moreover, a growing body of literature pointed toward nonsocial behavioral measures, such as motor skills, restricted and repetitive interests, sensory and visual processing, and attention disengagement [8,9,10,11]. Significant findings emerged from studies focusing on early motor development of children later diagnosed with NDDs [10,11]. Indeed, both fine and gross motor impairments are associated with NDD occurrence in the general population [8] and high-risk infants (i.e., siblings of children with a diagnosis of ASD, preterm and low birth weight infants) [9,10,12,13]. Overall, the most frequently reported early motor signs of NDDs are abnormalities in fluency, complexity, and variability of spontaneous general movements, and delays in early gross motor milestones [14,15], motor maturity [8,16], and motor functioning [17,18]. Findings reflect the heterogeneity of clinical symptomatology of NDDs, and instruments adopted to evaluate infants’ motor patterns are extremely variable, ranging from analysis of General Movements (GMs) based on the Prechtl method assessment to parental reports and specific clinical motor batteries [19,20,21,22,23,24,25].

The study of motor development at an early age may be crucial in ASD research fields [26,27]. Indeed, several studies showed that the motor patterns of infants later diagnosed with ASD appeared consistently less developed than in neurotypical peers (see review [28,29]) and that spontaneous motor activity was impaired already in the first months of life [25,30]. Moreover, some studies addressing differences in motor milestones documented the presence of early repetitive behaviors in infants with ASD compared to ones with typical development or developmental disabilities, starting from six months of age [31,32,33].

Relevant findings emerged from studies regarding the early motor development of siblings of children with ASD (high-risk (HR) infants) who present an increased risk (~20%) of developing ASD compared to the general population [34]. Mounting research highlights that their motor developmental trajectories differ from those at low risk (LR; siblings of children without a diagnosis of ASD/NDD) at a mere six months of age [22,24,35,36,37,38,39,40]. A recent systematic review showed that fine motor competencies at six months predicted the clinical outcome [9] and language skills of siblings of children with ASD [22]. Moreover, very early repetitive/stereotypical behaviors and atypical body movements (6–12 months [41], 12 months [42], and from 18 months [43]) were reported to be associated with ASD outcome in HR infants, in line with the acknowledgement of restricted and repetitive behaviors as a core symptom of ASD.

Although the use of innovative technological instruments has increased over recent decades, limited impacts were observed on early screening and detection of ASD/NDDs [44]. Given the importance of further exploring early motor trajectories, more research should be promoted to implement technological tools to obtain objective measures of motor patterns and describe quantitative and qualitative features of infant movements. Over the last few decades, a set of innovative technologies, including computer vision and motion sensor-based approaches, as well as machine learning approaches, were implemented with the aim to emulate GM analysis and pinpoint objective measures of kinematic analysis [45,46,47,48,49,50,51,52], but none were transferred to clinical practice.

Recently, the NIDA Network developed a motion-tracking technology named MOVIDEA [46]. The technology was developed under the arising need to identify an easy, cost-effective instrument to investigate early motor markers of NDDs. MOVIDEA is a noninvasive technology measuring quantitative motor patterns in infants through two- and three-dimensional kinematic analyses [46]. The current study describes the application of MOVIDEA on a large library of video recordings of spontaneous movements of LR and HR infants. Our aim was to detect early quantitative differences in the motor trajectories of HR infants who received a diagnosis of NDDs or who did not receive a diagnosis compared to LR age-matched peers. Due to the small sample size of HR infants who developed ASD (*n* = 3), we included them in the group of infants who developed NDDs (*n* = 15), resulting in a total sample size of 18 infants with NDDs.

Evaluation of kinematic parameters permitted the investigation motor differences as a function of age and outcome, with the aim to improve early discrimination and screening of infants at higher risk of developing NDDs.

## 2. Materials and Methods

### 2.1. Participants

The current study was performed within the NIDA Network, enrolling LR and HR infants with the aim of detecting early signs of ASD/NDD through the recording and analysis of infant cry and motor patterns at 10 days and 6, 12, 18, and 24 weeks of age. At 6, 12, 18, 24, and 36 months, each infant/toddler underwent a comprehensive clinical/diagnostic assessment using standardized tools/tests and structured interviews, with parents for checking the presence/absence of an ASD or NDD diagnosis. Infant recruitment and data collection are still ongoing. The present study was carried out according to the standards for good ethical practice and the guidelines of the Declaration of Helsinki. The study protocol was approved by the Ethics Committee of the Italian Institute of Health (Approval Number: Pre 469/2016). A written informed consent from a parent/guardian of each participant was obtained.

For the purposes of the current study, the sample included data from 103 infants (54 females and 49 males), of which 53 were LR (infants with a sibling without any clinical diagnosis and no family history of ASD) and 50 were HR (infants with an older sibling with a clinical diagnosis of ASD). The inclusion criteria for infants were: (1) gestational age ≥ 36 weeks; (2) birth weight ≥ 2500 g; (3) Apgar index > 7 at the 5th minute; (4) absence of known medical, genetic, or neurological conditions associated with ASD; (5) absence of major complications in pregnancy and/or delivery likely to affect brain development.

HR infants were subdivided into two groups based on their clinical outcome at 24 or 36 months (using the Autism Diagnostic Observational Scale [53] and the DSM-5 criteria), as confirmed by expert clinicians of the NIDA team blinded to experimental data, resulting in 18 HR infants diagnosed with NDDs (HR-NDD) and 32 who did not receive any diagnosis (HR-no diagnosis). Infants in the LR group did not receive any clinical diagnosis at 24 or 36 months.

Video recordings of spontaneous movements were performed at 10 days and 6, 12, 18, and 24 weeks of age at home while the infant was lying on the bed (either naked or in a bodysuit not covering the limbs). To acquire images of spontaneous movement of the full infant body, the camera was placed 50 cm above the child and at chest height. See Table 1 for the description of the participants’ characteristics and age during assessment.

### 2.2. Kinematic Analysis through MOVIDEA Software

Before applying MOVIDEA to the video library, each video was assessed for its eligibility for analysis. Then, one author cut each video to ensure the same properties, i.e., 3 min length, infant in supine position, in a condition of a quite awake state, well-being, and spontaneous motor activity, without crying or fussing episodes. Video frames containing interferences caused by the operator or parents or accidental movements of the camera were excluded from the analysis. Videos were analyzed by two coders blinded to the infant’s risk status (LR or HR). Prior to starting coding, coders were trained and reached an excellent degree of agreement.

The MOVIDEA software computed the motor features using two different approaches. First, the trajectories covered by the infant’s limbs during the free movement were extracted using a semi-automatic limb-tracking procedure. Second, movement quantification was performed through image processing techniques applied to the video frames. The software was developed using MATLAB ver. R2017a and its standard tools.

We analyzed the following set of kinematic features considered to eb meaningful for the identification of pathological motion patterns [46,54]:Quantity of motion (Qmean), i.e., the mean number of pixels where movement occurred divided by the total number of pixels in the image (percentage);Centroid of motion (C), i.e., the central point (2D coordinates measured in pixels) of the infant’s movement computed for each motion image extracted from the video recording (Figure 1), with analysis of standard deviation in the X (Cxsd) and Y directions (Cysd), velocity (Vcmean), and acceleration (Acmean);Periodicity for the left and right hands (H-periodicity) and for the left and right feet (F-periodicity), which were nondimensional parameters aimed at measuring the presence of repetitive movements in the motion of the limbs.

For details on the methods used to estimate measures as references and tracking methods, refer to Baccinelli and colleagues [46].

### 2.3. Statistical Analysis

An Analysis of Variance (ANOVA) with repeated measures was performed to analyze the kinematic parameters extracted by the MOVIDEA software (refer to Baccinelli and colleagues [46]), with the outcome (LR, HR-no diagnosis, and HR-NDD infants) as factor and age (10 days and 6, 12, 18, and 24 weeks) as the repeated measures. No significant sex differences were detected in the motor parameters analyzed. For this reason, data regarding sex were collapsed. Post-hoc comparisons were performed using Newman Keuls test. For all comparisons, significance was set at a *p*-value below 0.05. All statistical analyses were conducted using STATA 12.0 software (Stata Statistical Software: Release 12. College Station, TX, USA: StataCorp LP) [55,56].

## 3. Results

Using the MOVIDEA software, we evaluated specific kinematic parameters related to upper and lower limb movements in the space and using the head and the trunk as references (see Table 2).

### 3.1. Quantity of Motion

Analysis revealed a main effect of the outcome (F(2,319) = 8.83; *p* = 0.0002) with HR-NDD and HR-no diagnosis infants moving more than LR infants (*p* = 0.001 and *p* < 0.0001, respectively) and a main effect of the age (F(4,319) = 7.16; *p* < 0.0001) with differences at 10 days compared to 6, 12, 18, and 24 weeks (*p* < 0.0001 for all comparisons) (Figure 2).

Post-hoc comparisons performed on the two-way interaction outcome at x age (F(8,319) = 1.69; *p* = 0.1003) reported that HR-NDD and HR-no diagnosis infants moved more compared to LR infants at 12 weeks (*p* = 0.001 and *p* = 0.023, respectively). HR-no diagnosis infants moved more compared to LR infants at 18 weeks (*p* < 0.0001), and HR-NDD infants moved more compared to LR at 24 weeks (*p* = 0.031).

### 3.2. Centroid of Motion (Cxsd and Cysd)

Analysis revealed a main effect of the outcome (Cxsd: (F(2,319) = 10.76; *p* < 0.0001; Cysd (F(2,319) = 4.81; *p* < 0.0088. HR-NDD and HR-no diagnosis infants showed lower mean values of Cxsd compared to LR infants (respectively: *p* = 0.001; *p* < 0.001) (Figure 3) and a higher mean value of Cysd compared to LR infants (respectively: *p* = 0.027; *p* = 0.003) (Figure 4). Analysis revealed a main effect of age for Cxsd (F(4,319) = 3.50; *p* = 0.0081), with differences at 10 days compared to 6 weeks (*p* = 0.025) and 12 weeks (*p* = 0.033), at 6 weeks compared to 18 weeks (*p* = 0.026) and 24 weeks (*p* = 0.026), and at 12 weeks compared to 18 weeks (*p* = 0.035) and 24 weeks (*p* = 0.034). Analysis revealed a main effect of age for Cysd (F(4,319) = 2.39; *p* = 0.0508), with differences at 10 days compared to 6 weeks (*p* = 0.002), 12 weeks (*p* < 0.0001), 18 weeks (*p* = 0.009), and 24 weeks (*p* = 0.013).

Post-hoc comparisons performed on the two-way interaction outcome at x age (Cxsd: F(8,319) = 0.67; *p* = 0.7198; Cysd: F(8,319) = 0.25; *p* = 0.9812) revealed that HR-NDD infants showed a lower mean value of Cxsd in comparison to LR infants at 12 weeks (*p* = 0.004). HR-no diagnosis infants showed a lower mean value of Cxmean in comparison to LR infants at 10 days (*p* = 0.019), at 6 weeks (*p* = 0.043), at 12 weeks (*p* = 0.002), and at 18 weeks (*p* = 0.046). HR-No diagnosis infants showed a higher mean value of Cysd in comparison to LR infants at 18 weeks (*p* = 0.027).

### 3.3. Velocity of Centroid of Motion

Analysis revealed a main effect of the outcome (F(2,319) = 10.96; *p* < 0.0001), with HR-NDD and HR-no diagnosis infants showing reduced mean values of velocity of centroid of motion than LR infants (*p* < 0.0001 for both comparison groups) (Figure 5), and a main effect of age (F(4,319) = 3.22; *p* = 0.0130), with differences at 10 days compared to 12 weeks (*p* = 0.005), 18 weeks (*p* < 0.0001), and 24 weeks (*p* = 0.041), and at 6 weeks compared to 12 weeks (*p* = 0.040) and 18 weeks (*p* = 0.001).

Post-hoc comparisons performed on the two-way interaction outcome at x age (F(8,319) = 0.72, *p* = 0.6749) revealed that HR-NDD and HR-no diagnosis infants showed reduced mean values of velocity of centroid of motion compared to LR infants at 12 weeks (*p* = 0.001 and *p* = 0.003, respectively) and HR-no diagnosis infants at 18 weeks compared to LR infants (*p* = 0.008).

### 3.4. Acceleration of Centroid of Motion

Analysis revealed a main effect of the outcome (F(2,319) = 3.25; *p* = 0.0402), with HR-NDD infants showing increased acceleration of centroid of motion than LR infants (*p* = 0.050), and a main effect of age (F(4,319) = 5.88; *p* < 0.0001), with differences at 10 days compared to 12 weeks (*p* = 0.040) and 24 weeks (*p* = 0.009).

Post-hoc comparisons performed on the two-way interaction outcome at x age (F(8,218) = 4.55; *p* < 0.0001) revealed that HR-NDD infants significantly differed from both LR and HR-no diagnosis infants at 10 days, showing a higher mean value of acceleration of centroid of motion (*p* < 0.0001 for both comparison groups) (Figure 6).

### 3.5. Periodicity

Analysis did not reveal a main effect of the outcome for either hand periodicity (F(2,319) = 1.28; *p* = 0.2803) or foot periodicity (F(2,319) = 1.22; *p* = 0.2956). Analysis revealed a main effect of age (hand periodicity: F(4,319) = 6.39; *p* = 0.0001; foot periodicity: F(4,319 = 10.66; *p* < 0.0001). Hand periodicity was decreased at 10 days compared to 6, 12, 18, and 24 weeks (*p* < 0.0001) and at 18 compared to 24 weeks (*p* = 0.041) (Figure 7). Foot periodicity was decreased at 10 days compared to 6, 12, 18, and 24 weeks (*p* < 0.0001) and at 6 compared to 12 weeks (*p* = 0.038). By contrast, foot periodicity was increased at 12 weeks compared to 18 weeks (*p* = 0.003) and 24 weeks (*p* = 0.011) (Figure 8).

## 4. Discussion

The study of early motor patterns might be effective in predicting abnormal developmental trajectories of infants at risk for developing ASD/NDD. Technological tools able to obtain objective measures of specific motor parameters might support the early detection of NDDs. To this aim, the Italian Network for early detection of Autism Spectrum Disorder (NIDA Network) developed a noninvasive motion-tracking technology, named MOVIDEA, to measure quantitative motor patterns in infants at risk for NDDs.

The main objective of this study was to investigate very early motor trajectories of infants at risk for developing ASD through MOVIDEA, a semi-automatic software able to analyze 2D and 3D videos and provide objective kinematic features of their movements. To our knowledge, this is the first research devoted to a detailed characterization of kinematic motor patterns in HR infants within the first six months of life. Overall, our findings revealed that early motor patterns of HR infants later diagnosed with NDDs (HR-NDD) differed from those of HR and LR infants who did not receive any diagnosis (HR-no diagnosis: high-risk infants who did not receive any diagnosis; LR: low-risk infants with typical development).

First, HR infants later diagnosed with NDDs moved more compared to typically developing LR infants and differences also emerged between HR infants who did not receive any diagnosis and LR infants at 12 and 18 weeks. Thus, HR infants seem to exhibit more active motor profiles during the first six months of life. One possible explanation may be their different abilities to perceive and respond to external stimuli and generate adequate motor responses to them. Abnormalities in the inhibition/excitation system in the cortical and subcortical motor regions could underlie general motor dysfunctions associated with NDDs [57]. Self-regulation behaviors (general irritability, range of states, state regulation, regulation capability) were described as predictors of psychomotor development at 4 and 12 months [58]. Self-regulation difficulties were reported to be present in children with ASD at only one year of age [59,60]. Moreover, neurobiological studies showed that self-regulation in ASD is related to dysfunction in certain brain circuits associated with social–emotional processing [59,61], suggesting the importance of assessing neonatal motor features during development [58].

Second, variability of the “centroid of motion” parameter was different between HR and LR infants. Since the centroid of motion is the central point of the infant’s movement in a given motion image, its standard deviation in the X and Y directions throughout the video recording provide information regarding the variability of the location of the free movement’s central point. On the other hand, the velocity and acceleration of the centroid of motion provide a measure regarding how fast the global movement changes over time and if significant jerks of movements are present. Our data reported that HR infants later diagnosed with NDDs moved in the X direction with reduced variability at 12 weeks of age at the same time point during which they expressed an increased quantity of motion compared to LR infants. Also, HR infants who did not receive any diagnosis moved in the X direction with reduced variability across several time points of development (from 10 days to 18 weeks of age), and increased variability of centroid of motion was measured in the vertical direction (Cysd) at 18 weeks of age. Differences also emerged in the acceleration and velocity of centroid of motion. HR infants later diagnosed with NDDs showed increased acceleration of centroid of motion at 10 days after birth, compared to HR and LR infants who did not receive any diagnosis. Also, HR infants later diagnosed with NDDs and HR who did not receive any diagnosis demonstrated decreased velocity of centroid of motion compared to LR infants at 12 weeks of age. Differences remained significant at 18 weeks for HR infants who did not receive any diagnosis. Even if MOVIDEA provided mixed findings, it seems that the movement of HR-NDD infants is characterized by reduced variability and velocity as well as increased acceleration. It is possible that abnormal patterns of centroid of motion in HR infants may reflect abnormal quality of movement, as already observed through the GM evaluation, and could be predictive of a later diagnosis of NDDs. Typical GMs are characterized by complexity and variation [62], supporting the hypothesis that motor variability and appropriate integration of sensorimotor information are expressions of typical motor development [63,64]. A lack of movement variability was mostly identified in infants who developed delays or disabilities [65,66,67] and abnormal GMs were reported in several infants with ASD and NDDs [25,30,68,69]. Moreover, a previous study measuring kinematic parameters using wearable sensors in 1–8-month-old infants detected different acceleration features between at-risk infants diagnosed with developmental delay or with no diagnosis and infants with typical development [70].

Finally, MOVIDEA detected changes across age of development in hand and foot periodic movements. Periodicity was described as the repetition of the same movement multiple times with high amplitude of the end effectors [54]. The presence of certain motor repetitive behaviors in early infancy is considered a necessary step for the development of voluntary purposeful movements and seems to have an adaptive role. Thus, these are considered to be indices of a typical psychomotor development [71]. However, increased frequency of repetitive movements was widely described in various NDDs [33,72], reflecting a continuum extending from typical to atypical development. We speculate that atypical periodicity, especially for the upper extremities, could interfere with later manual exploration or poor development of fine motor skills. During the fidgety period, the upper limb movement repertoire undergoes a maturational process, with the enrichment of motor patterns addressing the exploration of the body and environment, while the lower limb repertoire appears to be functionally limited [50].

In conclusion, our results revealed that HR infants who received a diagnosis of NDD showed an increased general motor activity associated with reduced variability and velocity, as well as increased acceleration of global movement in the space. Moreover, we reported patterns of increased periodicity of limbs, especially for the upper limbs, throughout the first 12 weeks of age. This developmental profile may reflect the motor difficulties of HR-NDD infants during the writhing and fidgety periods and provide a window into which it is possible to more deeply investigate the development of motor competences in high-risk infants.

The subtle motion analysis through MOVIDEA has several advantages and strengths. Nowadays, most movement assessment systems used for infants require external tracking equipment, i.e., special markers or wearable sensors to be located on the limbs, which are not easily adaptable to different recording settings [52]. Conversely, MOVIDEA gives the chance to obtain measurements via noninvasive assessment without any additional special devices that could interfere with spontaneous infant motor activity. It is important to consider that the use of noninvasive technology may well fit recordings in real-life settings and naturalistic neonatal environments (e.g., hospitals, pediatrician ambulatories, homes), thereby allowing detailed motion analysis. In particular, possible applications of the MOVIDEA tracking system may be on home-video segments, as well as on videos recorded in clinical settings to routinely monitor infants at neurodevelopmental risk. These videos may be recorded directly by parents/caregivers or professionals minimally trained to videotape infants, taking into account the experimental set-up described in the methodological section. Moreover, MOVIDEA collects measurable and quantifiable information not based on visual scoring of infant motor performances, completed by clinicians during well-child visits or trained operators. We believe that this computer-based analysis of infants’ movements may support and integrate the analysis of motor patterns of infants at risk of NDDs in research settings [73,74]. Further studies could explore whether the features extracted by MOVIDEA software correlate with the qualitative and quantitative analysis performed by GM experts.

Some limitations of this study should be discussed. First, only a small sample size of HR-NDD infants was collected. Due to this limitation, it was not possible to carry out statistical analysis according to each separate NDD. Second, the accuracy of the extracted MOVIDEA features still need to be implemented and validated by experts in the field.

Overall, this experimental study identified potential early behavioral indexes that are related to later diagnosis of NDDs and may be precursors of altered movements during goal-directed behaviors. Moreover, the findings highlight the importance of a longitudinal assessment of motor development for LR and HR infants. Expanding knowledge regarding typical and atypical motor competencies is useful to detect a developmental motor trajectory as early as possible in the first weeks of life. Further video analyses should be carried out using the MOVIDEA software to increase the potential value of this objective, reliable, and quantifiable technology to identify early motor deficits in infants at risk for NDD.

## Figures and Tables

**Figure 1 brainsci-10-00379-f001:**
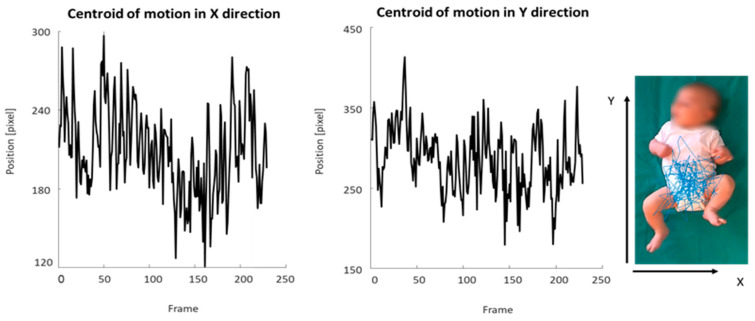
Graphic representation of the centroid of motion in the X and Y directions. The blue line depicts the trajectory of the centroid of motion of the infant’s movement extracted from a video recording.

**Figure 2 brainsci-10-00379-f002:**
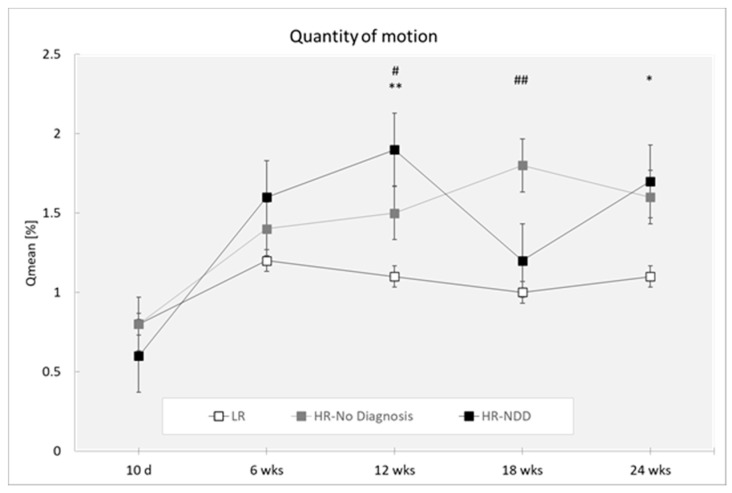
Quantity of motion in a 3-min session, measured at 10 days and 6, 12, 18, and 24 weeks in three groups of infants (LR, *n =* 53; HR-no diagnosis, *n =* 32; HR-NDD infants, *n =* 18). d = days; wks = weeks. Data are expressed as mean ± SEM. * *p* < 0.05 and ** *p* < 0.01 between HR-NDD and LR infants; # *p* < 0.05 and ## *p* < 0.01 between HR-no Diagnosis and LR infants.

**Figure 3 brainsci-10-00379-f003:**
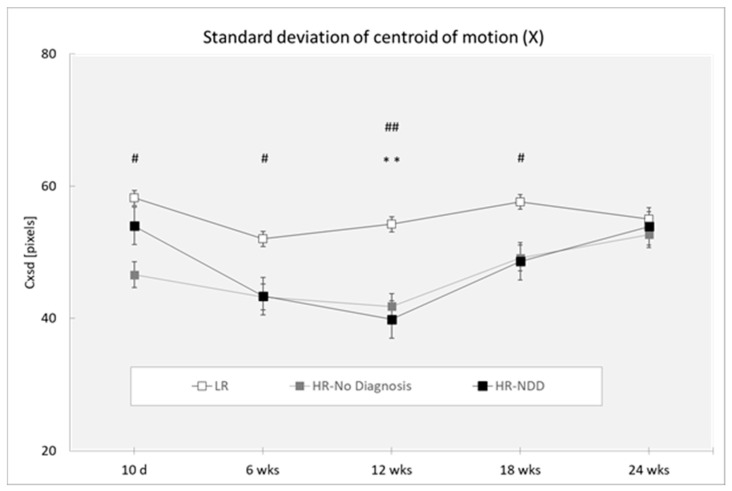
Standard deviation of centroid of motion in X direction (Cxsd) in a 3-min session, measured at 10 days and 6, 12, 18, and 24 weeks in three groups of infants (LR, *n =* 53; HR-no diagnosis, *n =* 32; HR-NDD infants, *n =* 18), d = days; wks = weeks. Data are expressed as mean ± SEM. ** *p* < 0.01 between HR-NDD and LR infants; # *p* < 0.05 and ## *p* < 0.01 between HR-no diagnosis and LR infants.

**Figure 4 brainsci-10-00379-f004:**
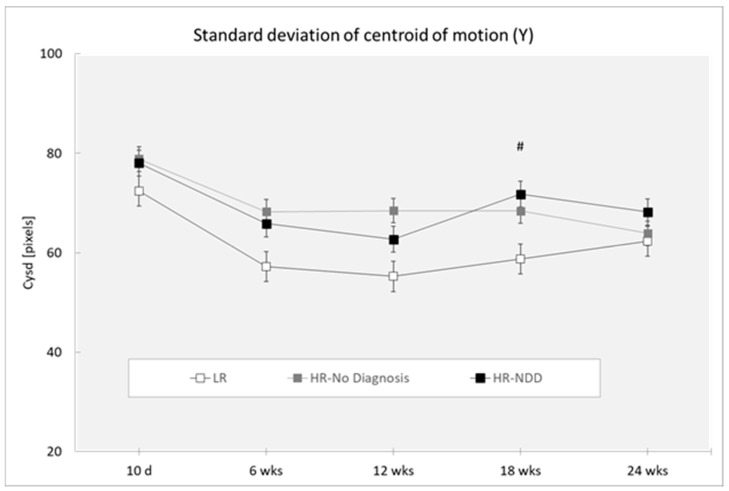
Standard deviation of centroid of motion in Y direction (Cxsd) in a 3-min session, measured at 10 days and 6, 12, 18, and 24 weeks in three groups of infants (LR, *n* = 53; HR-no diagnosis, *n* = 32; HR-NDD infants, *n* = 18). d = days; wks = weeks. Data are expressed as mean ± SEM. # *p* < 0.05 between HR-no diagnosis and LR infants.

**Figure 5 brainsci-10-00379-f005:**
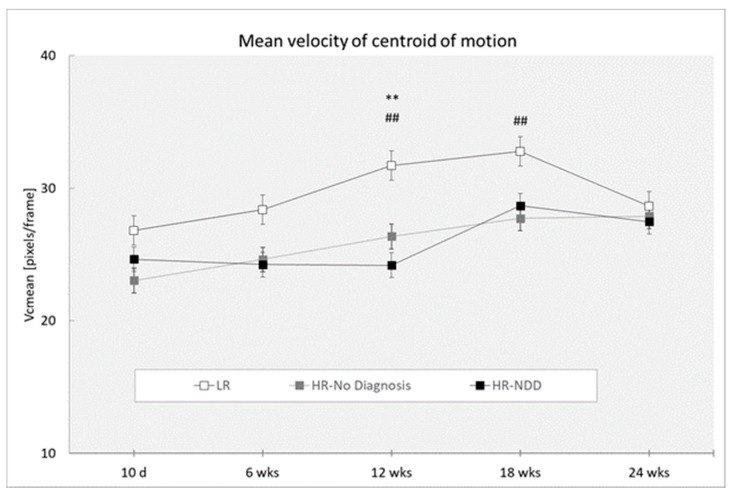
Mean velocity of centroid of motion in a 3-min session, measured at 10 days and 6, 12, 18, and 24 weeks in three groups of infants (LR, *n =* 53; HR-no diagnosis, *n =* 32; HR-NDD infants, *n =* 18). d = days; wks = weeks. Data are expressed as mean ± SEM. ** *p* < 0.01 between HR-NDD and LR infants; ## *p* < 0.01 between HR-no diagnosis and LR infants.

**Figure 6 brainsci-10-00379-f006:**
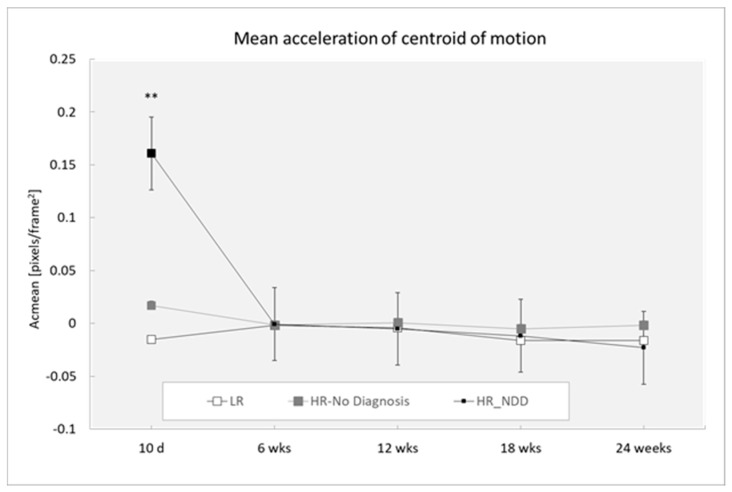
Mean acceleration of centroid of motion in a 3-min session, measured at 10 days and 6, 12, 18, and 24 weeks in three groups of infants (LR, *n =* 53; HR-no diagnosis, *n =* 32; HR-NDD infants, *n =* 18). d = days; wks = weeks. Data are expressed as mean ± SEM. ** *p* < 0.01 between HR-NDD and LR infants.

**Figure 7 brainsci-10-00379-f007:**
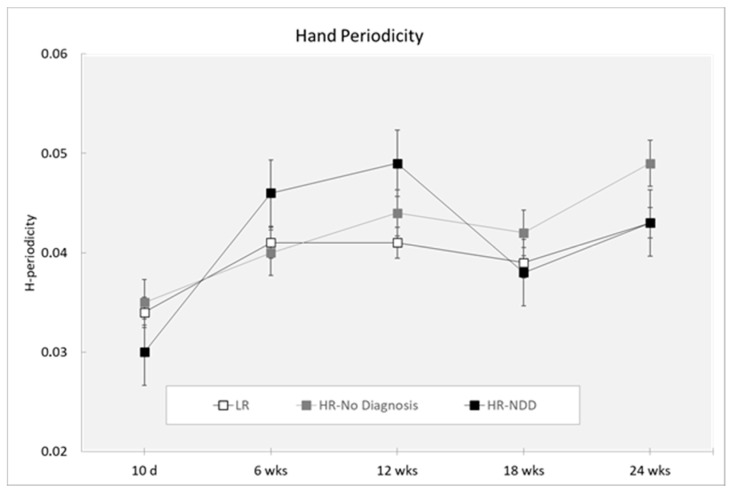
Hand periodicity in a 3-min session, measured at 10 days and 6, 12, 18, and 24 weeks in three groups of infants (LR, *n* = 53; HR-no diagnosis, *n* = 32; HR-NDD infants, *n* = 18). d = days; wks = weeks. Data are expressed as mean ± SEM. Periodicity is a nondimensional value.

**Figure 8 brainsci-10-00379-f008:**
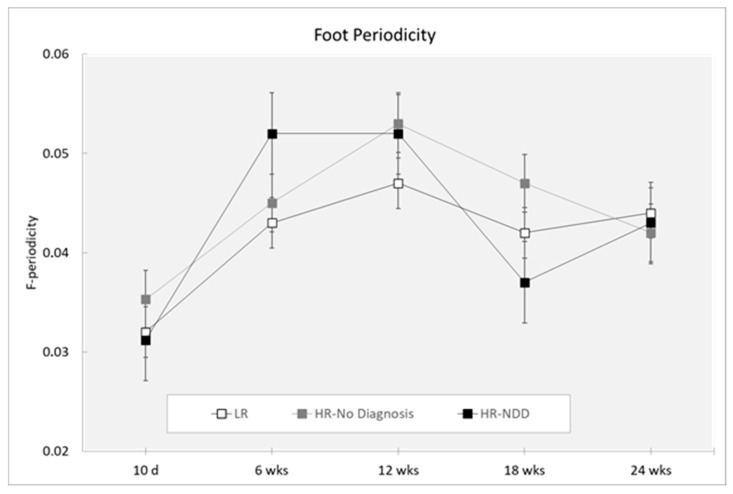
Foot periodicity in a 3-min session, measured at 10 days and 6, 12, 18, and 24 weeks in three groups of infants (LR, *n* = 53; HR-no diagnosis, *n* = 32; HR-NDD infants, *n* = 18). d = days; wks = weeks. Data are expressed as mean ± SEM. Periodicity is a nondimensional value.

**Table 1 brainsci-10-00379-t001:** Video recordings available at various ages of assessment.

Age of Assessment	LR (*n* = 53)		HR-No Diagnosis (*n* = 18)		HR-NDD (*n* = 32)	
	Female	Male	Female	Male	Female	Male
	*n*	*n*	*n*	*n*	*n*	*n*
10 days	16	31	7	15	5	7
6 weeks	21	43	12	21	4	12
12 weeks	24	46	15	24	5	13
18 weeks	16	36	15	23	5	8
24 weeks	12	28	8	13	7	14

LR: Low-risk infants with a typical development; HR-NDD: High-risk infants diagnosed with NDDs; HR-No diagnosis: High-risk infants who did not receive any diagnosis.

**Table 2 brainsci-10-00379-t002:** MOVIDEA kinematic parameters’ means (SD) of the three groups of infants (LR, HR-no diagnosis, and HR-NDD).

Parameter	LR		HR-No Diagnosis		HR-NDD	
	*n*	Mean (SD)	*n*	Mean (SD)	*n*	Mean (SD)
**Qmean**						
10 days	31	0.008 (0.006)	15	0.008 (0.003)	7	0.006 (0.005)
6 weeks	43	0.012 (0.008)	21	0.014 (0.005)	12	0.016 (0.009)
12 weeks	46	0.011 (0.006)	24	0.015 (0.009)	13	0.019 (0.005)
18 weeks	36	0.010 (0.007)	23	0.018 (0.011)	8	0.012 (0.005)
24 weeks	28	0.011 (0.009)	13	0.016 (0.009)	14	0.017 (0.009)
**Cxsd**						
10 days	31	58.210 (21.422)	15	46.589 (13.116)	7	53.976 (13.967)
6 weeks	43	51.999 (13.722)	21	43.243 (10.278)	12	43.372 (9.371)
12 weeks	46	54.223 (14.314)	24	41.800 (10.001)	13	39.843 (9.384)
18 weeks	36	57.612 (16.725)	23	49.113 (23.310)	8	48.621 (15.325)
24 weeks	28	55.010 (15.419)	13	52.645 (19.299)	14	53.890 (16.267)
**Cysd**						
10 days	31	72.398 (31.810)	15	78.819 (34.812)	7	78.012 (23.567)
6 weeks	43	57.223 (19.110)	21	68.222 (23.111)	12	65.830 (21.765)
12 weeks	46	55.230 (20.111)	24	68.450 (20.701)	13	62.692 (18.991)
18 weeks	36	58.716 (25.416)	23	68.412 (24.411)	8	71.768 (26.802)
24 weeks	28	62.320 (18.799)	13	63.899 (23.789)	14	68.175 (16.267)
**Acmean**						
10 days	31	−0.015 (0.071)	15	0.017 (0.052)	7	0.161 (0.307)
6 weeks	43	−0.002 (0.024)	21	−0.002 (0.028)	12	−0.001 (0.043)
12 weeks	46	−0.004 (0.030)	24	0.001 (0.027)	13	−0.005 (0.015)
18 weeks	36	0.016 (0.052)	23	−0.005 (0.033)	8	−0.011 (0.034)
24 weeks	28	−0.017 (0.139)	13	−0.002 (0.041)	14	−0.023 (0.030)
**Vcmean**						
10 days	31	26.835 (7.432)	15	23.040 (5.128)	7	24.640 (5.665)
6 weeks	43	28.396 (6.225)	21	24.624 (5.356)	12	24.250 (4.036)
12 weeks	46	31.726 (7.301)	24	26.374 (5.170)	13	24.194 (4.570)
18 weeks	36	32.792 (8.828)	23	27.730 (9.116)	8	28.687 (6.638)
24 weeks	28	28.652 (7.402)	13	27.884 (10.032)	14	27.466 (8.335)
**H-periodicity**						
10 days	31	0.034 (0.012)	15	0.035 (0.006)	7	0.030 (0.004)
6 weeks	43	0.041 (0.011)	21	0.040 (0.013)	12	0.046 (0.014)
12 weeks	46	0.041 (0.010)	24	0.044 (0.018)	13	0.049 (0.013)
18 weeks	36	0.039 (0.010)	23	0.042 (0.018)	8	0.038 (0.010)
24 weeks	28	0.043 (0.013)	13	0.049 (0.023)	14	0.043 (0.011)
**F-periodicity**						
10 days	31	0.032 (0.007)	15	0.035 (0.007)	7	0.031 (0.003)
6 weeks	43	0.043 (0.011)	21	0.045 (0.014)	12	0.052 (0.018)
12 weeks	46	0.047 (0.015)	24	0.053 (0.022)	13	0.052 (0.017)
18 weeks	36	0.042 (0.015)	23	0.047 (0.016)	8	0.037 (0.010)
24 weeks	28	0.044 (0.014)	13	0.042 (0.012)	14	0.043 (0.010)

LR: Low-risk infants with typical development; HR-no diagnosis: High-risk infants who did not receive any diagnosis; HR-NDD: High-risk infants diagnosed with NDDs. Qmean: Quantity of motion; Cxsd: Standard deviation in the X direction of centroid of motion; Cysd: Standard deviation in the Y direction of centroid of motion; Acmean: Mean acceleration of centroid of motion; Vcmean: Mean velocity of centroid of motion; H-periodicity: Periodicity for the left and right hands; F-periodicity: Periodicity for the left and right feet.

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
