# Peer review of "Early Motor Development Predicts Clinical Outcomes of Siblings at High-Risk for Autism: Insight from an Innovative Motion-Tracking Technology"

_brainsci, 2020, doi:10.3390/brainsci10060379_

Round 1

Reviewer 1 Report

The paper investigates the early motor trajectories of infants at risk for developing ASD. Authors used a semi-automatic software to analyse 2D and 3D infants’ movement videos.

The paper is structured well and motivates a well-defined clinical and research question related to ASD. The method and evaluation seem scientifically sound. The paper is structured and written well. All challenges and possible future works have been detailed in the paper.

Author Response

Point 1. The paper investigates the early motor trajectories of infants at risk for developing ASD. Authors used a semi-automatic software to analyse 2D and 3D infants’ movement videos. The paper is structured well and motivates a well-defined clinical and research question related to ASD. The method and evaluation seem scientifically sound. The paper is structured and written well. All challenges and possible future works have been detailed in the paper.

Response 1: Thank you for the positive evaluation given to the manuscript. We have really appreciated the valuable comment.

Reviewer 2 Report

Thank you for asking me to review this article. The identification of a measure to detect early signs that may lead to a diagnosis of autism spectrum disorder is an important area of research. This paper describes one such system that analyzes early motor movements as a possible biomarker for later ASD diagnosis. The authors present data which demonstrates that the MOVIDEA software does detect differences between HR infants who receive a NDD diagnosis versus HR infants who do not receive an NDD diagnosis and LR infants. This is an important first step to the development of a biomarker identification system. Results from the lab are important; however, the authors should discuss how this technology might be used outside of high-structured videotaped examples such as those used in this study. A brief discussion of how MOVIDEA might be used with family-generated video segments with the extra "noise" that was eliminated in their sample is needed to show how this system can be incorporated into real world practices.

Author Response

Point 1: Results from the lab are important; however, the authors should discuss how this technology might be used outside of high-structured videotaped examples such as those used in this study. A brief discussion of how MOVIDEA might be used with family-generated video segments with the extra "noise" that was eliminated in their sample is needed to show how this system can be incorporated into real world practices.

Response 1: Thank you for the valuable comment. We have highlighted this aspect in the Discussion section (Line 352) by including the statement: “It is important to consider that the use of non-invasive technology may well fit recordings in real-life settings and naturalistic neonatal environments (e.g., hospitals, pediatrician ambulatories, homes), allowing a detailed motion analysis. In particular, possible applications of MOVIDEA tracking system may be on home-video segments, as well as on videos recorded in clinical settings to routinely monitor infants at neurodevelopmental risk. Those videos may be potentially recorded directly by parents/caregivers or professionals minimally trained to videotape infants, keeping into account the experimental set-up as described in the methodological section.”